# Polish Patients’ Needs and Opinions about the Implementation of Pharmaceutical Care in Diabetes

**DOI:** 10.3390/ijerph20020945

**Published:** 2023-01-04

**Authors:** Magdalena Waszyk-Nowaczyk, Weronika Guzenda, Karolina Kamasa, Łucja Zielińska-Tomczak, Magdalena Cerbin-Koczorowska, Michał Michalak, Piotr Przymuszała, Beata Plewka

**Affiliations:** 1Pharmacy Practice Division, Chair and Department of Pharmaceutical Technology, Poznan University of Medical Sciences, 6 Grunwaldzka Street, 60-780 Poznan, Poland; 2Student’s Pharmaceutical Care Group, Pharmacy Practice Division, Department of Pharmaceutical Technology, Poznan University of Medical Sciences, 6 Grunwaldzka Street, 60-780 Poznan, Poland; 3Department of Medical Education, Poznan University of Medical Sciences, 7 Rokietnicka Street., 60-806 Poznan, Poland; 4Chair and Department of Computer Science and Statistics, Poznan University of Medical Sciences, 7 Rokietnicka Street, 60-806 Poznan, Poland

**Keywords:** pharmaceutical care, patient education, procedural instruction, diabetes, pharmacists, community pharmacy

## Abstract

The study aimed to get to know patients’ opinions on implementing pharmaceutical care for diabetic patients in a community pharmacy to prevent and effectively and holistically approach the treatment of people with diabetes. It was based on an authorial survey form and conducted from August to October 2021 in a community pharmacy in Poznan, Poland. A total of 131 pharmacy patients over 18 years were included in the study. Results showed that the vast majority of patients confirmed their interest in pharmaceutical care in diabetes conducted by pharmacists. Moreover, 79.4% of respondents would like to benefit from medicines use review, while 87.0% confirmed an interest in the ‘New Drug’ service, with diabetic patients being particularly interested in this (*p* = 0.2447). Most respondents were also interested in education on how to use a glucose meter, administer insulin and use a lancing device. In addition, the study showed patients’ insufficient knowledge about risk factors and prevention of diabetes with the need for patient education. As the source of funding, 91.7% of diabetic patients indicated the National Health Fund. Given that such a service has not been implemented in Poland yet, this study may support established teams at the Supreme Pharmaceutical Chamber or the Ministry of Health in introducing such new services.

## 1. Introduction

Among healthcare workers, pharmacists are one of the most available professions as community pharmacies are widely accessible and often the first line of contact for patients seeking help with health-related issues [1]. Research conducted in Great Britain shows that 89% of its residents can reach a pharmacy within 20 min [2]. The waiting time for an appointment with a physician is sometimes too long for patients, which is why they very often use the services of pharmacists due to the easy access to professional knowledge and the ability to quickly obtain medical advice. In fact, patients contact pharmacists up to ten times more often than general practitioners [3]. The signing of the Act on the Pharmacy Profession on 17 December 2020, resulted in paying special attention to the role of pharmacists in the Polish healthcare system, allowing to take full advantage of their knowledge and potential. One of the main assumptions of this regulation is the introduction of pharmaceutical care and, therefore, different services for patients to improve the health indicators of Poles [4]. The concept of pharmaceutical care in Poland was described in a recent report on the comprehensive analysis of its implementation process [5]. Among the planned activities, we can distinguish pharmaceutical services, drug reviews, the New Drug service, minor ailments care program, continued prescription, cardiovascular disease prevention program, and administering vaccinations in pharmacies (for better visualization presented as Figure 1) [5]. Moreover, as a professional group, pharmacists can also contribute to promoting a healthy lifestyle or monitoring pharmacotherapy [6]. There is a need to use the potential of pharmacists in caring for patients with chronic diseases. As studies show, the services provided by pharmacists in pharmacies are effective in the treatment of diabetes [7]. Randomized studies show that pharmacists’ actions lead to key savings and improve cost efficiency and cost–benefit analysis compared to a health system that does not use pharmacists’ potential. Improving treatment outcomes can, therefore, significantly relieve strained healthcare systems [8].

Meanwhile, diabetes mellitus is a common and still not fully understood health problem affecting people worldwide, especially in developed countries. There has been a continuous increase in the number of cases of the disease in recent years [9]. Diabetes mellitus is also one of four non-communicable diseases that has been recognized by the World Health Organization as an epidemic of the 21st century. Moreover, a continuous increase in the incidence is forecasted in the oncoming years, which is why early diabetes prevention and appropriate control of patients are so important [10]. In 2017, more than 425 million people worldwide had diabetes [11]. It is estimated that the number of patients in 2035 will increase to 592 million [12]. In 2017, diabetes was one of the leading causes of death among U.S. residents. Reportedly, every 6 s, one person dies due to diabetes and its complications [13]. In Poland, in 2018, the number of patients was 2.9 million (1.3 million adult men and 1.6 million adult women), i.e., every 11th inhabitant of Poland was affected [14]. Type 2 diabetes is the most common form of diabetes, and in highly developed countries, approximately 87.0% to 91.0% of all people with diabetes have type 2 diabetes [7]. The incidence of diabetes is increasing, which is mainly explained by the influence of environmental factors. The most important is abdominal obesity. Statistics show that 50.0% of diabetics are obese (BMI > 30 kg/m^2^), and 90.0% are overweight (BMI > 25 kg/m^2^). The risk of developing diabetes also increases with age. The incidence peak occurs among the inhabitants of developing countries between the ages of 40 and 45 and in developed countries over 60 [9]. A healthy diet, reduction of obesity, ceasing to use stimulants, and correction of lipid disorders are effective methods for the improvement of disturbed carbohydrate metabolism [15]. Screening tests are also an important element of preventive medicine, enabling early identification of health problems among patients. Polish Diabetological Association recommends carrying out such examinations using the oral glucose tolerance test or fasting blood glucose levels [16]. Effective therapeutic tools, interdisciplinary patient care, and increasing the availability of diagnostics seem crucial in controlling the escalation of the diabetes epidemic [17].

Pharmacists can play an important role in the integrated care of a diabetic patient. Consultation with pharmacists improves the overall health performance of diabetics in relation to ordinary care [18]. Research conducted in Nigeria shows that patients covered by pharmaceutical care services achieved higher health-related quality of life than those in standard health care [19]. Their role may begin with the identification of high-risk patients and directing them to the appropriate medical unit for further comprehensive diagnostics. It is also necessary to educate the patient, assess compliance with therapeutic recommendations, or obtain clinically relevant information during pharmaceutical consultations [20]. Nutritional counseling is an important part of diabetes prevention. As an obesity reduction program offered by Scottish pharmacies showed, such an action can contribute to a few percent weight loss among patients. Moreover, program participants reported satisfaction with the conducted services [21]. One of the goals of pharmaceutical care in diabetes is also proper pharmacotherapy, which is crucial in the treatment of chronic civilization diseases. Defining clinical drug problems, identifying drug interactions during medicines use reviews, or deepening patients’ knowledge about the disease may significantly support the treatment process [22]. Early identification of potential drug problems may prevent them from becoming real problems with a negative impact on the patient’s life and health [23]. Pharmacists can also verify the correctness of blood glucose meter use by patients and increase their awareness of hygiene issues related to blood glucose measurements [24]. Moreover, in Poland, 1⁄3 of people suffering from diabetes still remain undiagnosed, and the importance of early diagnosis and prevention of diabetes is still too rarely discussed. Meanwhile, screening for diabetes in most cases is possible thanks to simple and cheap tests, such as blood glucose levels [16]. Performing such a test in community pharmacies seems a good solution because of patients’ better access to pharmacies than to laboratories or doctors [3]. Implementing such modifications in the Polish healthcare system would also relieve other medical staff members, including nurses and doctors [25].

As demonstrated, pharmaceutical care as a health service consisting of the integrated cooperation of a pharmacist and a patient brings many benefits, which is why it is so important to include it as a standard procedure in Poland and other countries. However, it should be remembered that, after all, the main beneficiaries of pharmaceutical care are patients. Therefore, it is necessary to take a closer look at their level of awareness and learn about their expectations regarding the additional services provided in community pharmacies. In addition, getting to know the opinions of patients about additional services provided by employees of pharmacies, including pharmaceutical care in diabetes, may facilitate the development of this field and the implementation of such solutions in the healthcare system. Pharmaceutical care services for diabetic patients and groups at high risk of disease have not yet been introduced to Polish pharmacies, which is why it is so important to obtain information on the way society views these services and their usefulness, as well as the needs and expectations of patients. Consequently, the aim of the study was to evaluate patients’ level of awareness of prevention and selected risk factors for the development of diabetes (as potential areas of education by pharmacists) and their opinions on pharmaceutical care in diabetes in a community pharmacy.

## 2. Materials and Methods

### 2.1. Study Design

The study was conducted among patients of the ‘Medicover’ community pharmacy in Poznan (Poland) between August and October 2021 using an authorial questionnaire form. The inclusion criteria for the study were pharmacy patients’ voluntary consent to participate and the age of 18 years or older. The exclusion criterion was lack of consent or resignation from participation in the study and conditions in which logical contact with the patient could not be established. The study was approved by the Bioethical Committee at the Poznan University of Medical Sciences (Case number: 833/21).

### 2.2. The Survey Questionnaire and Procedure

The survey consisted of three parts and is presented in Appendix A. Part A contained questions about general respondent information such as their gender, age, height, and weight to calculate BMI, place of residence, and education. Part B was composed of questions related to diabetes, such as prevention, risk factors, self-assessment of own knowledge, and family history of the disease. It also allowed patients to answer questions on their opinions and interest in pharmaceutical care in diabetes, including education by a pharmacist on insulin administration, lancing device operation, ‘New Drug’ service, information about the risk factors for diabetes, or provision of medicines use reviews in community pharmacies. The plan for introducing the ‘New Drug’ service was included in the above-mentioned report on pharmaceutical care as a type of pharmaceutical consultation accompanying the introduction of a new medicinal product to the patient, which they have never used before, for example, in the case of a new diagnosis or a change in the treatment. The report recommends financing the service from public funds. Part C was intended for diabetic patients and involved questions on respondents’ potential interest in pharmaceutical care services in diabetology provided by a pharmacist, private visits in the field of pharmaceutical care, and opinions on the financing of such services. Prior to the actual patient survey, a pilot study involving ten patients was performed to check the comprehensibility of survey forms [26]. Additionally, patients had the opportunity to ask questions when filling out the form. Before respondents started filling in the questionnaire, they were informed about the purpose of the study, its anonymous nature, as well as the scientific purpose of gathering the information. A secluded location at the pharmacy was provided for respondents to ensure their comfort during the study.

### 2.3. Data Analysis

The collected data were analyzed using Microsoft Excel, STATISTICA 12 (StatSoft), and STATA14 StataCorp LLC software programs. The analysis process involved the verification of differences among the respondents in terms of age, gender, place of residence, education level, height, weight, and the occurrence of risk factors for diabetes. In the case of nominal variables, the chi-square test of independence was used, and for observed small or zero values, Fisher’s exact test was used. For interval variables with no consistency with the normal distribution and for the data from the ordinal scale, the Mann–Whitney U test was used to compare the obtained results between the two groups. In case of simultaneous comparisons between more than two groups, the Kruskal–Wallis test was used with Dunn’s post hoc tests. All tests were analyzed at the significance level of a = 0.05 and were considered statistically significant at *p* < 0.05.

## 3. Results

### 3.1. Participants’ Characteristics

A total of 131 patients participated in the study, including 79 women and 52 men. The most numerous age group of respondents were patients aged 18 to 29 (29.0%) and 30 to 39 years (28.2%). The majority of respondents had higher education (70.2%). Most patients (72.5%) had BMI within the norm. Significantly more male participants of the study (36.5%) were overweight than females (16.5%) (*p* = 0.007). It was also noticed that the body weight and risk of being overweight among patients increased with age (*p* = 0.004). Moreover, 55.0% of patients had a family history of diabetes. The largest percentage of patients were seeing only one physician (32.8%), while 22.9% were not under the supervision of any specialist. Detailed characteristics of the study group are presented in Table 1. Briefly, 40.5% of the respondents confirmed the presence of chronic diseases. The most frequently diagnosed diseases among patients were hypothyroidism (25.2%) and hypertension (19.1%), followed by irritable bowel syndrome and mental illnesses. The incidence of diabetes among the respondents was 9.2%. Detailed data are presented in Figure 2.

### 3.2. Patients’ Knowledge of Diabetes and Its Prevention

The study intended to investigate patients’ self-assessment of their knowledge of diabetes. Results showed that 38.2% of respondents regarded their knowledge about diabetes as sufficient, while 45.0% as insufficient, and 16.8% did not know. Inhabitants of cities statistically more often noticed a lack in their knowledge, saying it was insufficient (50.5%) in relation to the rural population (27.6%), as shown in Figure 3 (*p* = 0.011).

Patients who considered their knowledge about diabetes as sufficient were significantly older than those who assessed it as insufficient (Figure 4, *p* < 0.001).

The vast majority of respondents (95.4%) agreed that diabetes prevention is important. When asked about the methods of preventing the development of diabetes, 86.3% of patients knew that regular physical activity is one of the preventive measures. Interestingly, the average age and BMI values of patients who agreed with that statement were significantly lower (Figure 5, *p* = 0.005). Moreover, 79.4% of patients agreed that the lack of physical activity might contribute to the occurrence of type 2 diabetes.

When asked about factors contributing to type 2 diabetes, 96.9% knew that being underweight was not a risk factor—Among them, 61.4% were women, and 38.6% were men. The differences in terms of gender were significant (*p* < 0.001). Similarly, the vast majority of patients (93.9%) knew that being overweight is a contributing factor to type 2 diabetes, of which most were women (61.8%) and the differences between them were also significant (*p* < 0.001). Moreover, 91.6% of the respondents considered correct eating habits as a preventive factor for diabetes, with bigger awareness in this regard demonstrated by younger patients and those achieving lower mean BMI values (Figure 6, *p* = 0.046). Among patients with diabetes, 75.0% knew that proper eating habits might have a preventive effect on the development of diabetes (*p* = 0.027).

Moreover, patients who previously declared insufficient knowledge about the disease were statistically more often unaware of it (Figure 7, *p* = 0.003).

Briefly, 62.6% of respondents were aware of the positive impact of a diet based on products with a low glycemic index in preventing the development of diabetes. The awareness increased with the patient’s education level (Figure 8, *p* = 0.046).

Most patients (96.9%) knew that a diet with high glycemic index products does not prevent the development of diabetes, and this view was significantly more common among chronic disease patients (*p* < 0.001). The vast majority (90.8%) of patients were aware that eating large amounts of fruit was not a preventive factor for the development of the disease, and these were most commonly men (*p* = 0.047). Only 36.6% of the respondents knew that smoking could influence the development of type 2 diabetes, and these were more often women (*p* < 0.001). In addition, chronic disease patients were more aware of this, as healthy patients accounted for 61.4% of the unaware respondents (*p* < 0.001). Compared to 60.8% in the case of women, only 46.2% of men knew that the presence of hypertension might favor the development of diabetes (*p* = 0.029). Among patients without diabetes in the family, only 47.1% said that hypertension might contribute to its development, compared to 63.9% of patients with a family history of diabetes (*p* = 0.037).

As the survey showed, 80.2% of respondents had had their blood glucose levels measured in the past. This was significantly more common in patients with chronic diseases (Figure 9, *p* = 0.010). Patients without chronic diseases accounted for 80.8% of the respondents who never had their glucose levels measured.

### 3.3. Patients’ Opinions on Pharmaceutical Care Services with Particular Emphasis on Diabetology

Most (81.7%) of the surveyed patients would like to be informed by pharmacists on risk factors for developing diabetes. Willingness for such education was not expressed by 7.6% of them, and 10.7% answered ‘I don’t know’.

In the case of the ‘New Drug’ service, 87.0% of the respondents showed a willingness to use this service provided by pharmacists, 3.8% were not interested, and 9.2% answered ‘I don’t know’. Patients with a diabetes family history were significantly more interested in this service (*p* = 0.030). Moreover, 79.4% of patients would like to benefit from medicines use reviews provided by a pharmacist (associated, among other things, with the detection of possible drug problems or interactions and dosage control), 5.3% were not interested, and 15.3% did not know. Respondents who were interested in the ‘New Drug’ service were also more inclined to receiving the medicines use review (*p* < 0.001). Patients with a family history of diabetes were also significantly more interested in such a service (*p* = 0.024). Statistically significant differences in patients’ preferences were presented in Table 2.

Moreover, the vast majority of patients (84.7%) would like a pharmacist to educate them in the field of glucose meter use. Such a solution was not of interest to 6.9% of respondents, and 8.4% answered they did not know. There were also statistically significant differences in respondents’ answers to this question in terms of age, as shown in Figure 10.

Additionally, respondents who indicated that the prevention of diabetes was important were significantly more interested in education about glucose meter use conducted by the pharmacist (*p* = 0.028). Moreover, 84.7% of respondents also showed an interest in education on insulin administration or the use of the lancing device conducted by a pharmacist, while 3.8% of them were not interested, and 11.5% did not know. Respondents who indicated the need for such training were also significantly more interested in pharmacist-led medicines-use reviews (*p* < 0.001). Statistically significant differences in terms of patients’ preferences for procedural instruction services were presented in Table 3.

Among patients with diabetes, 66.7% of respondents showed interest in pharmaceutical care services in diabetology conducted by a pharmacist in a community pharmacy, while 33.3% answered ‘I don’t know’. Interest in such services was more common among women, as shown in Figure 11 (*p* = 0.029).

The vast majority of diabetic patients (91.7%) believed that the coverage of pharmaceutical care should be financed by the National Health Fund [Narodowy Fundusz Zdrowia—Polish national agency financing health services for insured persons financed by compulsory health insurance]. In the meantime, 33.3% of patients would also be interested in private (paid by the patient) pharmaceutical care appointments in the field of diabetology, 16.7% would not be interested, and 50.0% had no opinion.

## 4. Discussion

Pharmaceutical care has not been legally sanctioned in Poland for a long time, and what is related to it, these services were not commonly available to Polish patients. It is worth noting that the inclusion of pharmaceutical care services into the daily duties of pharmacists is beneficial not only for patients but also for pharmacists [27]. Getting to know patients’ views on pharmaceutical care in diabetes can help understand how patients perceive these services, as well as their expectations of future interventions. Thanks to this, the implemented solutions can contribute to an even greater improvement in the health condition of patients [28]. Pharmacists, as representatives of the profession of public trust, can significantly influence the improvement of the quality of life of patients. Our survey showed that nearly one-third of the respondents were seeing one doctor, and 22.9% were not under the supervision of any doctor. Some diabetic patients also have comorbidities [29]. Pharmacotherapy in such cases is complex and requires taking several medications simultaneously leading to a threat of multi-drug use. According to the report of the National Health Fund, over 9% of Poles over 65 years of age take more than five drugs a day [30]. This creates the risk of drug problems such as drug side effects or drug interactions. Given their availability in community pharmacies, this can serve as another example of why pharmacists should expand their pro-health activities, as pharmaceutical care is a service that, among others, ensures the safe and rational use of drugs among patients. Searching for and solving drug-related problems are the basic tasks of a pharmacist, as defined by Hepler and Stranda [31].

According to the new regulations in force from 17 December 2020, pharmaceutical care as a health service means performing drug reviews along with the assessment of pharmacotherapy, identification of individual drug problems, developing a personalized pharmaceutical care plan, or performing diagnostic tests. Medicines use review is a service that significantly reduces the risk of adverse events in patients [32]. Moreover, 79.4% of the respondents in this study indicated that they were interested in having a pharmacist perform a medicine use review. This is consistent with another research carried out in Poland in 2020, which showed the interest in such a service provided by a pharmacist at a level of over 70% [33]. On the other hand, the United States is an example of a country where such services are widely available and enjoy great interest. American patients find them valuable, and satisfaction with the quality of the services provided is high [34]. Similarly, 87% of our respondents were interested in the ‘New Drug’ service—a form of pharmaceutical consultation for patients whose treatment has been modified or a new disease has been identified [5]. Realizing the sense of the pharmacotherapy and clarifying doubts is important in patients’ complying with it [35]. A study carried out in Pakistan showed that over 96.0% of patients were willing to take advantage of a pharmaceutical consultation, thus confirming the leading role of a pharmacist in the community [28]. Examples from other countries also show that a pharmacist plays a key role in the process of optimal pharmacotherapy. Avoiding drug discrepancies at a level of 52.0% is possible by educating patients about drug compatibility. Moreover, it can result in savings allowing the reduction of treatment costs by over 43.0% [32].

However, pharmaceutical care focuses not only on the correctness of pharmacotherapy of patients but also on the assessment of the risk of diseases, especially civilization-related ones. Preventive measures are also an important element of the Individual Pharmaceutical Care Plan. Moreover, the implementation of such procedures in Poland could contribute to savings of PLN 4 billion, thus allowing for huge savings for the state budget [36]. According to the National Health Fund report, published on 13 November 2019, on the eve of World Diabetes Day, every 11th Polish resident is burdened with this disease [37]. Diabetes mellitus is also one of four non-communicable diseases recognized by the World Health Organization as the epidemic of the 21st century [10]. Problems with carbohydrate metabolism are growing at an alarming rate, which is why it is so important to effectively use healthcare professionals for the prevention and an adequate therapeutic process. The study showed that body weight and the risk of being overweight tended to increase with patients’ age. This is in line with a study in the United States, where the percentage of obese men aged 20–39 was 40.3%, and among those aged 40–59, it was 46.4% [38]. Aging is associated with an increased likelihood of abdominal obesity, a major factor increasing insulin resistance and the risk of metabolic syndrome. Moreover, age is a non-modifiable factor that increases the risk of developing diabetes. As the 2019 report shows, in Poland, at least 1/4 of people aged 70 or older suffer from at least one disease: diabetes or coronary heart disease [39]. Most respondents were aware that being overweight and insufficient physical activity are potential risk factors in the development of diabetes. However, only 36.6% connected smoking cigarettes with an increased risk of developing this disease. These results are consistent with studies conducted in Hong Kong, where little awareness of the correlation between smoking and type 2 diabetes mellitus, its health complications, and misconceptions about quitting smoking were noted [40]. In the study, 55.0% of respondents considered the presence of hypertension as a factor in the development of diabetes. The clinical characteristics of Bangladeshi inhabitants present an identical problem—only half of them indicate hypertension as a factor in the development of diabetes [41]. In our study, only 38.2% of the respondents considered their knowledge of diabetes sufficient, and over 94.0% agreed that diabetes prevention is important. Meanwhile, knowledge of controlling and treating diabetes at the right time helps to reduce mortality and incidence [42]. Patient education in this area is still low, and they do not have sufficient knowledge about this disease. Many patients associate diabetes only with excessively high blood glucose levels without being aware of serious and dangerous complications that can be caused by chronic and untreated hyperglycemia [43]. For example, a study in India proved that only half of the respondents knew what diabetes is, and less than 27.0% of people were aware of complications such as microangiopathies or macroangiopathies [44]. Among our respondents, 81.7% agreed that pharmacists should educate patients on diabetes in community pharmacies. Therefore, there is a noticeable space for the use of pharmacists’ potential and, thus, the implementation of pharmaceutical care services in order to improve patients’ awareness [45]. Such activities can be extremely useful in the prevention and treatment of chronic diseases. Research carried out in Brazil presents interesting conclusions on the information provided by pharmacists—an improvement in the interest in complying with doctors’ recommendations was noticed, as well as a reduction in the number of drug problems among patients [46].

Moreover, only 45.0% of patients had their glucose levels tested in the last year. Meanwhile, as other studies show, 65.8% of the respondents would be willing to have blood glucose levels tested in a community pharmacy [33]. Pursuant to the Ordinance of the Minister of Health of 21 January 2022, on the list of diagnostic tests that may be performed by pharmacists, they are allowed to measure blood glucose after the completion of the appropriate qualification courses. The possibility to perform screening tests in pharmacies plays an important role in the early detection of problems, thus ensuring effective therapy, delaying the progression of hyperglycemia, and lowering the costs of therapy [47]. Pharmacists can also support the treatment process of diabetic patients by educating them about the correct technique of self-performed blood glucose measurements, interpreting results, timing of measurements, and taking appropriate behaviors depending on the obtained blood glucose result [48]. As community pharmacies are the most common places of choice for patients when it comes to purchasing blood glucose meters or lancing devices, pharmacists have great potential to educate patients on the topic [3]. In our study, 84.7% of respondents were interested in being educated by a pharmacist on the use of a blood glucose meter. The same percentage of patients would like to receive help from a pharmacist in handling the lancing device or administering insulin. Interestingly, patients who were not interested in this form of education had a higher mean age than those interested. This means that more attention should be paid to addressing and educating the elderly in order to increase their awareness. Meanwhile, there are studies proving that even a one-time consultation in a community pharmacy drastically reduces the probability of errors during the measurement of glucose levels. Only 17.0% of patients who consulted a pharmacist made mistakes compared to 59.0% who did not use such a service [49]. In another study, among patients who visited pharmacists to learn how to measure their blood glucose correctly, the percentage of those who had doubts about the test was reduced by half. Moreover, 80% of diabetics indicated a pharmacy as the appropriate place for such services and rated the program as needed [50].

Most diabetic patients would like pharmaceutical care services in diabetes. Among them, 91.7% indicated the National Health Fund as the body that should finance them, while 33.3% would also be willing to participate in private (paid) visits. A similar observation was made in Saudi Arabia, where 29% of respondents expressed a willingness to pay an additional fee for pharmaceutical consultations in community pharmacies [51]. The aging society and staff shortages result in limited options for comprehensive treatment. Doctors do not have time to meet all the therapeutic needs of patients during visits [52]. This is a particular obstacle when patients suffer from comorbidities and take several medications. As a result, involving pharmacists in the joint management of diabetes by developing a pharmaceutical care plan may contribute to the improvement of patients’ health. Regular consultations with a pharmacist regarding adequate blood glucose control and care for a diabetic patient translate into a reduction in the frequency of medical visits, thus increasing the productivity of doctors [53]. By involving pharmacists in improving diabetes management, patients would profit from health benefits, while other healthcare workers in hospitals would have more time to diagnose and treat more critical cases requiring more serious intervention [54].

We acknowledge the limitations of this study. It was a single-center study conducted in a big city. Therefore, the population of respondents should not be viewed as representative of the general Polish population for several reasons, including the residents of bigger cities being generally younger and also potentially with higher education levels than residents of rural areas, which was quite noticeable in our sample. Consequently, the views of patients in other parts of Poland may differ, and we believe there is a need to continue the study on a bigger sample with patients from different backgrounds. Moreover, only 12 patients in our sample had been diagnosed with diabetes. Therefore, it also seems beneficial to plan a qualitative study with diabetic patients on their needs and opinions regarding pharmaceutical care in order to reach deeper and more insightful data than is possible with quantitative methodology.

## 5. Conclusions

The study showed the interest of Polish patients in receiving pharmaceutical care services in diabetes (such as medicines use reviews, patient education on diabetes, insulin administration, and training in the use of glucose meters or lancing devices) and also proved the high need for them. For this reason, such services should be introduced to pharmacies as a standard. The healthcare system in Poland is overloaded, and the society is aging, which creates a need for better and broader use of the potential of pharmacists. Meanwhile, patients are open to services provided by pharmacists in community pharmacies, so the implementation of such changes has great potential. Examples from other countries show many advantages resulting from the inclusion of these activities into the healthcare program. Comprehensive supervision and care for the patient’s health, early and facilitated prophylactics and detection of health problems, or patient education may serve as examples. Furthermore, the introduction of the above services into the daily practice of pharmacists also contributes to a more positive perception of the pharmacist’s profession by society.

## Figures and Tables

**Figure 1 ijerph-20-00945-f001:**
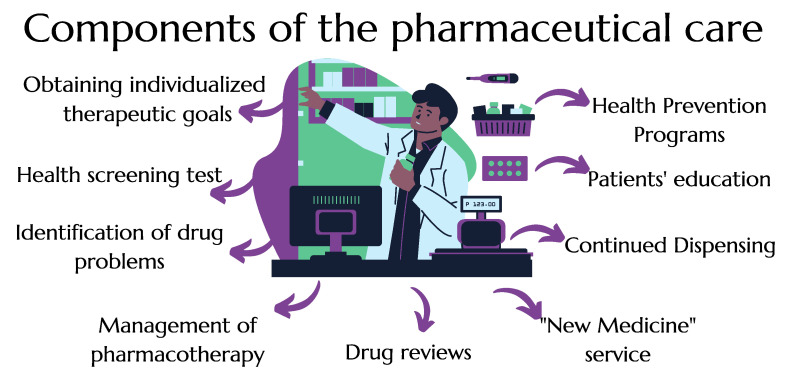
Proposed pharmaceutical services [5].

**Figure 2 ijerph-20-00945-f002:**
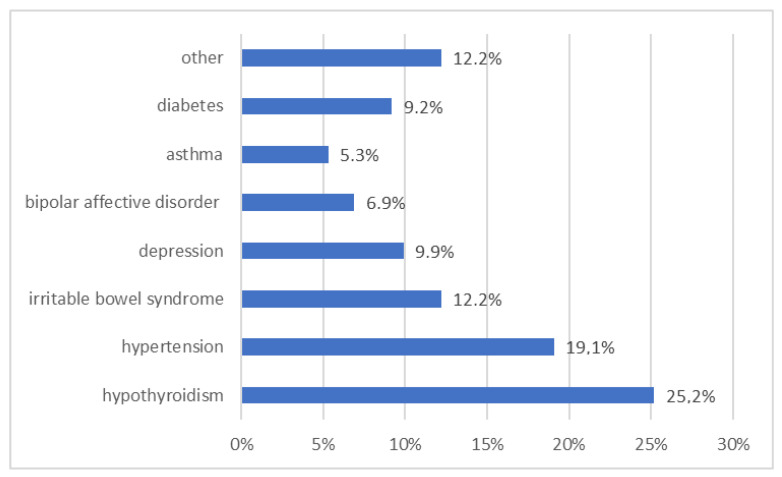
Diseases declared by the surveyed patients (*n* = 53).

**Figure 3 ijerph-20-00945-f003:**
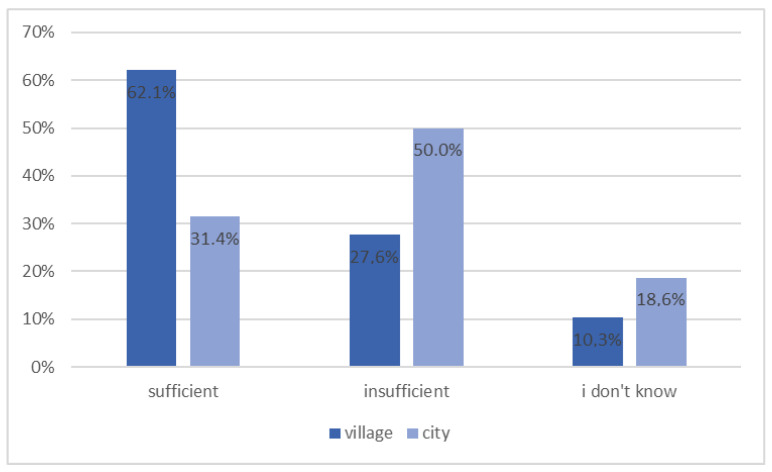
Patients’ self-assessment of their knowledge about diabetes depending on the place of residence (*n* = 131, *p* = 0.011).

**Figure 4 ijerph-20-00945-f004:**
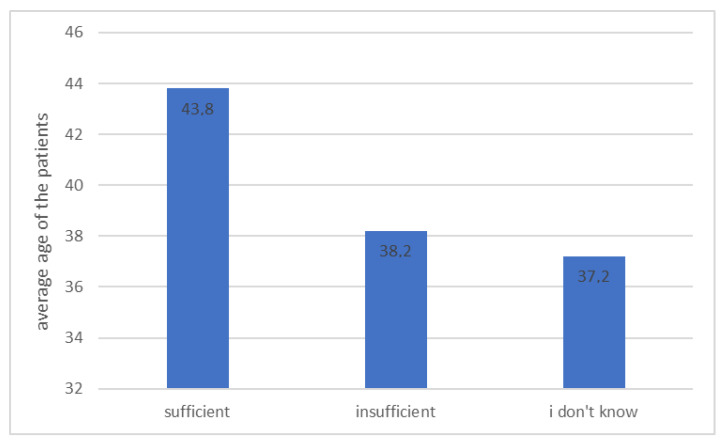
Average age of patients depending on their self-assessment of knowledge about diabetes (*n* = 131, *p* < 0.001).

**Figure 5 ijerph-20-00945-f005:**
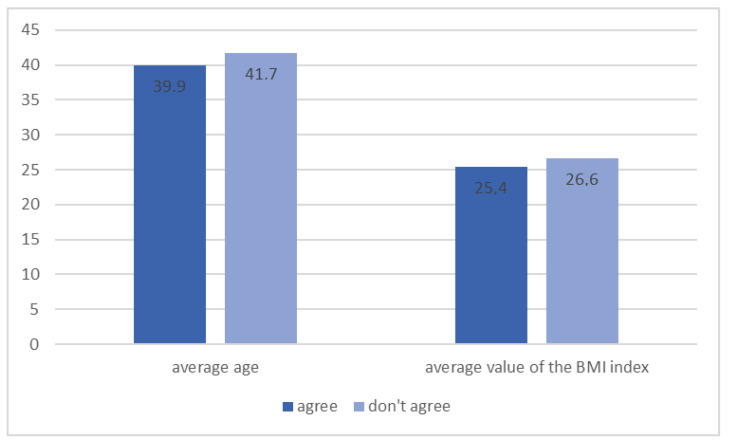
Patients’ opinion on the role of regular physical activity in preventing the development of diabetes depending on average age and BMI level (*n* = 131, *p* = 0.005).

**Figure 6 ijerph-20-00945-f006:**
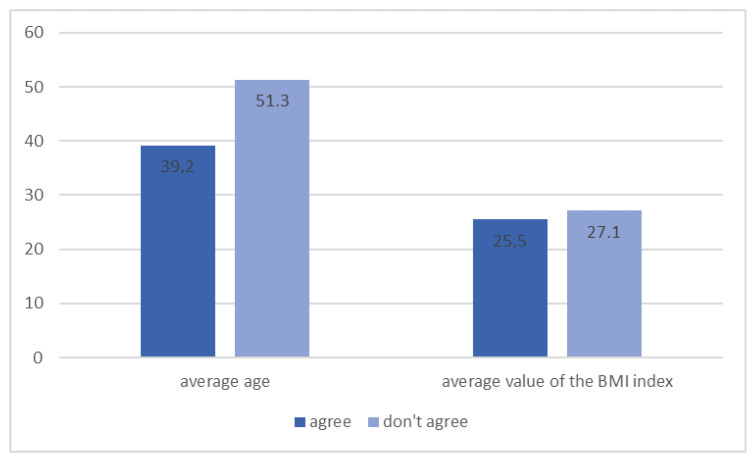
Patients’ opinion on proper eating habits in preventing the development of diabetes depending on the average age and BMI value (*n* = 131, *p* = 0.046).

**Figure 7 ijerph-20-00945-f007:**
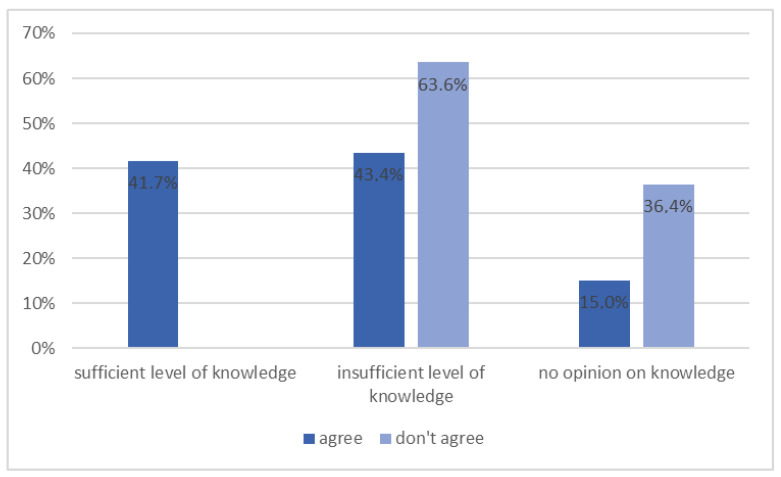
Patients’ opinion on proper eating habits in preventing the development of diabetes depending on their self-assessment of knowledge about diabetes (*n* = 131, *p* = 0.003).

**Figure 8 ijerph-20-00945-f008:**
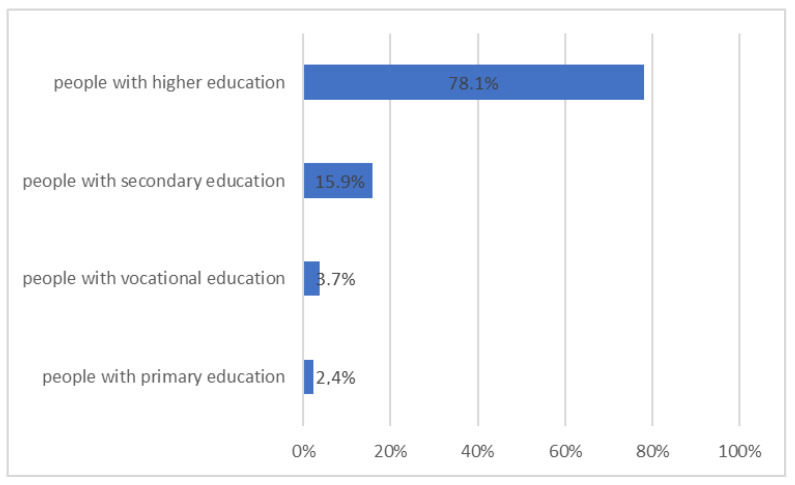
Patients’ awareness of the role of a diet based on products with a low glycemic index in the prevention of diabetes (*n* = 82, *p* = 0.046).

**Figure 9 ijerph-20-00945-f009:**
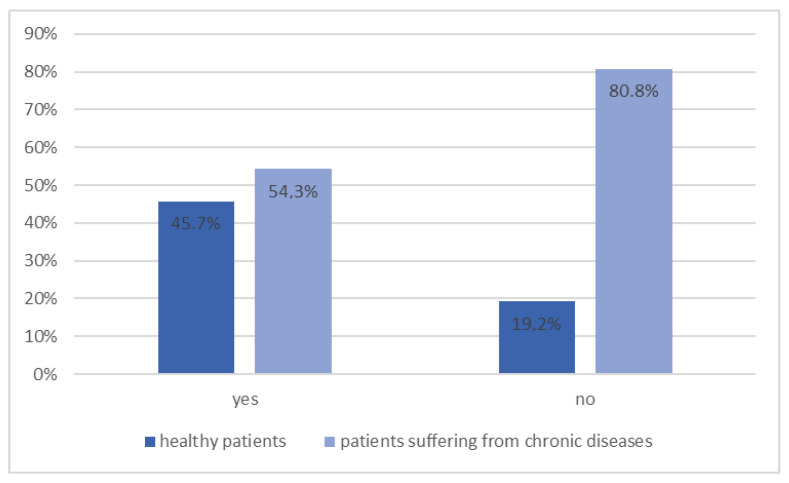
Distribution of chronic diseases among patients depending on whether their blood glucose level has ever been tested (*n* = 131, *p* = 0.010).

**Figure 10 ijerph-20-00945-f010:**
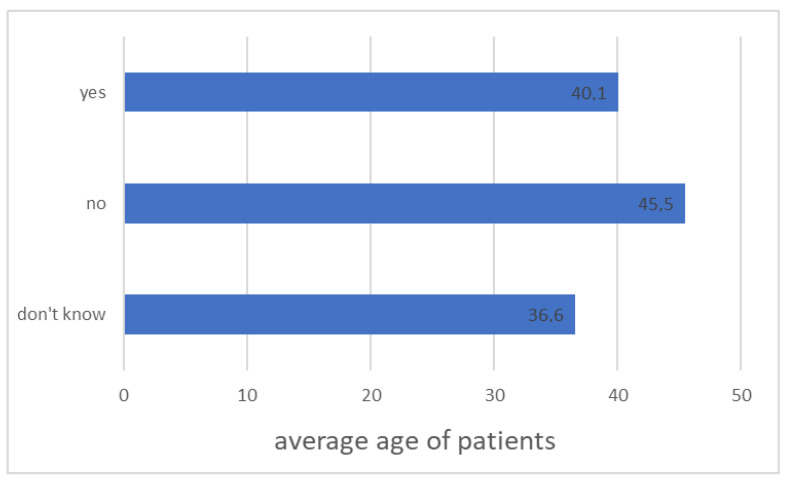
Average age of patients by their opinions on the education in the glucose meter use by a pharmacist (*n* = 131, *p* < 0.001).

**Figure 11 ijerph-20-00945-f011:**
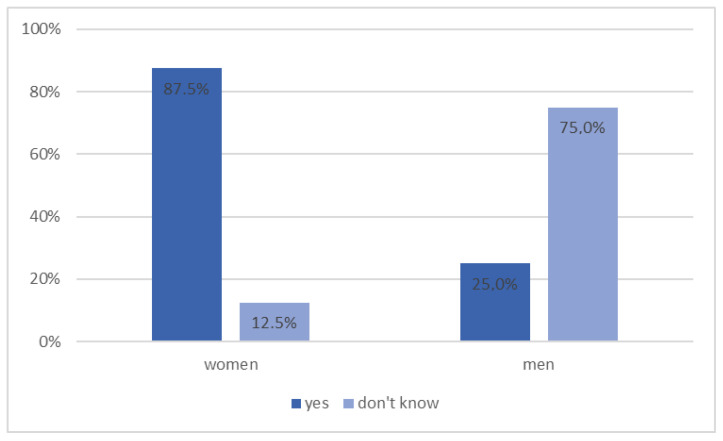
Diabetic patients’ interest in pharmaceutical care services in diabetology by gender (*n* = 12, *p* = 0.029).

**Table 1 ijerph-20-00945-t001:** Characteristics of the study group.

Gender	Female	60.3%
	Male	39.7%
Age range	18–29	29.0%
	30–39	28.2%
	40–49	15.3%
	50–59	14.5%
	60–69	6.9%
	70–79	6.1%
Education level	Primary	2.3%
	Lower secondary	1.5%
	Secondary	22.1%
	Vocational	3.8%
	Higher	70.2%
BMI [norm: 18.5–24.9]	Below the norm	3.1%
	Within the norm	72.5%
	Above the norm	24.4%
Diabetes in family	Yes	55.0%
	No	38.9%
	I don’t know	6.1%
Number of physicians attended	1 physician	32.8%
	2 physicians	16.0%
	3 physicians	17.6%
	4 physicians	6.1%
	More than 4 physicians	4.6%
	No physicians	22.9%

**Table 2 ijerph-20-00945-t002:** Patients’ interest in drug-related services—only statistically significant differences between subgroups.

Questions about New Services Posed to Respondents	Patients’ Subgroups	Yes	No	I Don’t Know
Would you like a pharmacist to provide you with information on a newly prescribed drug (New Drug service)?	Family diabetes burden	91.7%	4.2%	4.2%
	No family history of diabetes	86.0%	4.0%	10.0%
	*p*-value	*p* = 0.030		
Would you like a pharmacist to review your medication?	Patients interested in the New Drug Service	86.8%	3.5%	9.7%
	Patients not interested in the New Drug Service	20.0%	60.0%	20.0%
	Patients with no opinion	27.3%	0.0%	72.7%
	*p*-value	*p* < 0.001		
Would you like a pharmacist to review your medication?	Family diabetes burden	86.1%	4.2%	9.7%
	No family history of diabetes	76.0%	6.0%	18.0%
	*p*-value	*p* = 0.024		

**Table 3 ijerph-20-00945-t003:** Patients’ interest in procedural instruction services—only statistically significant differences between subgroups.

Questions about New Services Posed to Respondents	Patients’ Subgroups	Yes	No	I Don’t Know
Would you like the pharmacist to educate you in the use of the glucose meter?	Patients believing that diabetes prevention is important	97.3%	88.9%	80.0%
	Patients with no opinion on the prevention of diabetes	2.7%	11.1%	20.0%
	*p*-value	*p* = 0.028		
Would you like the pharmacist to educate you on insulin administration/lancing device operation?	Patients interested in a drug review	85.8%	60.0%	40.0%
	Patients not interested in a drug review	3.6%	40.0%	6.7%
	Patients with no opinion	10.9%	0.0%	53.3%
	*p*-value	*p* < 0.001		

## Data Availability

The data used to support the findings in this study are available from the corresponding author upon request.

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
