# Peer review of "Polish Patients’ Needs and Opinions about the Implementation of Pharmaceutical Care in Diabetes"

_ijerph, 2023, doi:10.3390/ijerph20020945_

Round 1
Reviewer 1 Report
Dear authors,
The manuscript submitted by Ms. M. Waszyk-Nowaczyk and her colleagues present an interesting study regarding the providing of pharmaceutical services for diabetes patients.
Please find my suggestions and comments below:
1. The exact period should be presented in Abstract and Material and methods section.
2. Pharmacists are considered the most accessible healthcare professionals and a valuable human resource. I think this issue should be added. (please see: Ahmad, A.; Alkharfy, K.M.; Alrabiah, Z.; Alhossan, A. Saudi Arabia, pharmacists and COVID-19 pandemic. J. Pharm. Policy Pract. 2020, 13, 1–3)
3. A short description of pharmaceutical care programs in Poland should be provided.
4. Also, a short description of diabetes programs / protocol in Poland should be added.
5. Lines 72-74: more recently data should be added (please see: https://www.mdpi.com/1660-4601/17/12/4456)
6. Line 91: Some information regarding the insulin resistance should be added. Subsequently, you can continue to present the information on improving of the tissue sensitivity to insulin.
7. Introduction section should be reorganized. It is too extensive.
8. Line 177: Authors should add information regarding the target group.
9. What were the exclusion criteria? Inclusion criteria should be detailed.
10. It would be better if the questionnaire is included as supplementary material
11. The “New Drug” service should be detailed. It is an implemented service ? By whom? What is it? Who finance it?, etc.
12. Line 190: More details regarding the pilot study must be added.
13. Table 1 should be reorganized. The data should be separated by lines.
14. Lines 280-281: The formulation is inappropriate.
15. Line 314 is not appropriate for Results section. Idem Lines 339-341.
16. Lines 369-381: n=9. In my opinion this question is irrelevant. I suggest deleted.
17. Line 392-394: which survey is it about?
18. Line 398: Drug incompatibility is irrelevant for community pharmacies. I suggest delete this aspect.
19. Line 514: In my opinion "despite quite a big sample" is not appropriate for this sample. Moreover, the number of diabetic patients was very low. This can be another limitation of the study.
20. It would be better if authors present a set with the proposed pharmaceutical services.
Author Response
Dear Reviewer, thank you for reviewing our paper and the provided suggestions. We implemented them as disclosed below.
Reviewer 2 Report
· The introduction section is too long and it is better to be shorter and clear.
· The method section can be clearer for better understanding of the discussion results.
· Please bring the questionnaire as a supplement file
· Please bring analysis of Reliability and Validity of a Questionnaire results.
· Too much images were presented in the manuscript, that’s may be confusion for the readers. Is it possible to bring data in a table?
· Please present the results section in a clearer form
Author Response

(The authors gave the same response as above.)

Reviewer 3 Report
Thank you for the opportunity to read this manuscript and congratulations to the authors for their work. I highly appreciate the text submitted for review.
Here are some suggestions for improvement:
Review the aims of the study: "the aim of the study was to evaluate patients’ opinions on pharmaceutical care in diabetes in a community pharmacy" as... "the study also analyzed patients’ knowledge of prevention and risk factors for the development of diabetes to determine the potential areas of patient education by pharmacists."
Review the numbers of the sample size (e.g. Figure 5 and 6).
Review the format and edit some of the figures (e.g. Figure 21, 22,...)
Review some of the references.

Author Response

(The authors gave the same response as above.)

Round 2
Reviewer 1 Report
Thank you for all your corrections.
Reviewer 2 Report
Authors responded my comments. My decision is to accept in the present form.
Regards,